# Relationship between Muscle Tone of the Erector Spinae and the Concave and Convex Sides of Spinal Curvature in Low-Grade Scoliosis among Children

**DOI:** 10.3390/children8121168

**Published:** 2021-12-10

**Authors:** Jacek Wilczyński

**Affiliations:** Posturology Laboratory, Collegium Medicum, Jan Kochanowski University, 25-369 Kielce, Poland; jwilczynski@onet.pl; Tel.: +48-603-703-926

**Keywords:** scoliosis and scoliotic posture in children, muscle tone of erector spinae, spinal curvature convexity and concavity

## Abstract

The objective of the present research was to assess the relationship between muscle tone of the erector spinae and the concave and convex sides of spinal curvature in low-grade scoliosis found among children. The study included 251 children, aged 7–8. Examination of the spine and body posture was carried out using the Diers Formetric III 4D optoelectronic method. Surface electromyography (sEMG) was used to assess erector spinae muscle tone. The trial was carried out using the 14-channel Noraxon TeleMyo DTS apparatus. The highest generalised tone (sEMG amplitude) of the erector spinae occurred in the case of scoliosis. The higher the angle of curvature, the greater the erector spinae muscle tone. Regardless of the position adopted during examination of the thoracic spine, greater erector spinae tone (sEMG amplitude) was exhibited on the convex side of the spinal curvature. However, in the area of the lumbar spine, greater tone (sEMG amplitude) of the erector spinae occurred on the curvature’s concave side. The exception was the test performed in a standing position, during which greater muscle tone was noted on the side of the convex curvature. In therapeutic practice, within the thoracic section, too tense erector spinae muscles should be stretched on the convex side of the scoliosis, while in the lumbar region, this should be performed on the concave side. However, each case of scoliosis requires individually tailored treatment. The current research has applicative value and does fill a research gap with regard to erector spinae muscle tone in young children experiencing low-grade scoliosis. The development of scoliosis is associated with asymmetry and an increase in erector spinae tone. The uneven distribution of its tone, occurring on both sides of the spine and in its various segments, causes destabilisation and its abnormal progression.

## 1. Introduction

The onset and development of scoliosis (IS) depend on both of the following factors: etiological as well as biomechanical [1]. Those first mentioned can be greatly varied and initiate the onset of scoliosis (IS). However, the second type is typical for all scoliosis types, and irrespective of etiology, they work in accordance with the laws of gravity and growth. The given factor is in control of scoliosis development [2]. Of significance is the force of gravity present along the long spinal axis [3,4]. In proper conditions, the spine maintains its physiological shape thanks to internal balance [5,6]. This balance is made up of passive and, at the same time, active elements. Properly shaped vertebrae, joined by joints as well as ligaments, undergo the compressive forces of gravity [7]. In the case of multi-curvatures, one of them is primary (curvatura primaria). A primary curvature is in the shape of an arch that is regular and equal in both of its halves. The primary curvature is always larger and the most permanent and, therefore, the least susceptible to correction. The primary curvature is, in most cases, compensated by a secondary curvature of the opposite direction. Secondary curvature (curvatura secudariae) is a symptom of striving to compensate for disturbances in the mechanical axis of the spine. While primary curvature is a negative factor, the secondary one, restoring balance and statics of the trunk, although under changed conditions, should be treated as a positive phenomenon. The attempt to compensate for disturbances in body axis in the curvature of the spine by producing counter-curvatures or distortions of the pelvis is called compensation [8]. Intervertebral discs, due to their flexibility and contraction, counteract the compressive force and evenly distribute loads over entire surfaces of the vertebrae However, the back muscles, due to coordinated action, ensure an active and balanced spine. A share of researchers finds the cause of scoliosis to be in bone structures, in the so-called passive apparatus of the spine. They highlight the significance of disturbances in the epiphysial cartilage of the spinal vertebrae [9]. Nonetheless, at present, the majority of researchers recognise the primary disturbances as being in the Central Nervous System (CNS), which lead to disturbances in muscle balance. The changes taking place in the passive spinal elements are, according to them, secondary in nature [10]. In scoliosis, muscle balance regarding the main and active spinal stabilisers, i.e., the erector spinae, can be subjected to disturbances caused by primary dysfunctions in the structures of the CNS [11]. This, as a result, leads to disturbances in the activity of the erector spinae and in its tension, causing curvatures to develop. Much allows to conclude that the discrete functional changes in the CNS are indeed the primary source of idiopathic scoliosis [12]. Under the influence of the etiological factor, disturbance of internal spinal balance occurs. During the first phase, only a narrowing occurs in the case of the intervertebral discs which are on the concave side of the scoliosis and a loss is visible in their elasticity. Simultaneously, an uneven increase in erector spinae muscle tone (sEMG amplitude) can be observed. Both in the diagnosis and in preventative treatment of scoliosis, much attention is devoted to the evaluation of differences in erector spinae tone between the concave and convex scoliosis sides and at different spine levels [13]. However, is the erector spinae always contracted on the concave side and stretched on the latter side of the curvature, regardless of its location? In scientific literature on the subject, we may note reports containing contradictory findings [14]. Determining asymmetry (differences) regarding erector spinae tone between the convex/concave sides, separately for the sections of the thoracic and lumbar spine sections, is of primary significance when selecting the proper treatment method for scoliosis [15]. The research objective of the present work was assessing the correlation between the muscle tone of erector spinae and the concave and convex sides of spinal curvature in low-grade scoliosis among children. It was assumed that a clear correlation is present between erector spinae muscle tone and the concave as well as convex sides of the spinal curvature in low-grade scoliosis found among children. In the thoracic spine, greater erector spinae tone occurs on the convex side of the curvature. In contrast, in the lumbar region, greater tone is noted on the concave side.

## 2. Material and Methods

The study participants comprised 251 individuals, children aged 7–8, totalling 113 (45.02%) girls and 138 (54.98%) boys. The study was performed at the Posturology Laboratory of Jan Kochanowski University in Kielce (Poland) in 2017. Prior to the trial, all of the subjects provided informed consent for inclusion in the trial. The study was performed in accordance with the regulations of the Declaration of Helsinki. The University Bioethics Committee, Jan Kochanowski University, Kielce (Resolution No. 5/2015), provided approval of the research protocol. The criteria for inclusion in the study were the following: presence of scoliosis (IS), scoliotic posture (PS), age between 7 and 8 years, parents’ or legal guardians’ written informed consent to perform the examination. The exclusion criteria for participation were syndromes related to the CNS and/or locomotor system preventing adequate psychomotor development, disorders potentially causing pathological posture: genetic syndromes, hormone-based disorders, neuromuscular diseases, congenital motor system defects, or lack of written consent to take part in the test protocol. The children and their parents/guardians were provided with information regarding the objective of the trial, on its course and duration. The selection of children aged 7–8 for the study was related to the fact that the first critical period of posturogenesis occurs in the initial school period. Between the ages of 7–8, the lifestyle of children changes. The essence of this change consists in the transition from a free (individually regulated movement by the child—effort and rest) to an imposed system comprising several hours of sitting, sometimes in inappropriate conditions. The second critical period of posturogenesis is related to the pubertal growth spurt that occurs in girls between the ages of 11–12, while in boys, between the ages of 13–14. The first scoliotic changes appear most often in these stages.

Examination of body posture and the spine was carried out via surface topography via raster stereography. Three-dimensional spinal analysis comprises a mix of optical imaging techniques and processing of digital data. This is a method used for quick and non-contact photogrammetric measurement which allows us to analyse the back and spine of a patient. The quick image transfer to a computer allows data to be analysed immediately following the test. At the time of conducting measurements, the patient adopts habitual posture. In order to test body posture as well as the spine, the DiCAM 2.4.9 DIERS Formetric 4D basic average program was used [16]. Body posture evaluation was performed 2-fold, with a 1-min interval in between the trials. The researcher, noting the results for the clinical trial, decides which is a most accurate reflection of the child’s actual habitual posture and this is the only trial subjected to further analysis. The body posture examination itself lasts approximately 15 min. In accordance with the guidelines provided by the producer of the Diers Formetric III 4D device, scoliosis and/or scoliotic posture are identified by taking the values obtained for 3 variables into consideration: pelvic tilt (/millimetres), lateral deviation (/millimetres), and surface rotation (/degrees). Scoliotic posture (PS) occurs in the case of pelvic tilt totalling 1–4 mm, lateral deviation between 1–4 mm, and surface equalling 1–4°. Scoliosis (IS) is present when pelvic tilt and lateral deviation equal or exceed 5 mm, while surface rotation totals or is greater than 5°. It was understood that neither scoliosis nor scoliotic posture were present in cases for which the 3 requirements stated above were not met [16]. After body posture examination, the children left the room and rested under supervision for approximately 1 h. During that time, the children could sit, draw, watch cartoons, or walk around; however, it was essential the children did not perform anything tiresome, e.g., running. 

Surface electromyography (sEMG) was implemented to evaluate erector muscle tone in the spine. The study was carried out using the 14-channel Noraxon TeleMyo DTS apparatus. The erector spinae sEMG amplitude was evaluated in the thoracic as well as lumbar sections on the left and right sides of the spine. The higher the sEMG amplitude, the greater the erector spinae muscle tone. Regel electrodes (30-mm diameter) were attached to the skin of the patients. At the application point, the skin was degreased and cleaned with an abrasive fluid in order to achieve resistance between the cleaned skin and the applied electrode, with an amplitude lower than 2 kΩ. These electrodes were placed in a position parallel to the direction of the evaluated muscle fibre. They were placed every 20 mm. The program responsible for signal modification and clearing the raw record of extreme, maximum, and minimum deviations was selected. Following this, the raw signal processing option was chosen for calculating the mean SEMG amplitude. The processed SEMG mean amplitude, given in millivolts (mV), was recorded. The test results took a voltage scale with a 100-millisecond time interval into account. The contrast track recording mode was applied for the testing procedure. The sEMG amplitude was recorded on both sides of the erector spinae, in the thoracic and lumbar segments, at the curvature arch apex and in the following positions: habitual standing; resting—prone (the knee joints are extended in the knee joints while the upper limbs are positioned alongside the trunk); in conditions of isometric contraction—prone position, the subject lifts his/her trunk within the lumbar spine region (approximately 30°) and maintains this position for a duration of 10 s, in prone position, upper trunk stabilised, the subject lifts both of the lower limbs within the range of hip joint mobility (approximately 15°) for 10 s. A total of 16 SEMG amplitude measurements were taken for each child: standing position, thoracic segment, left side; standing position, thoracic segment, right side; standing position lumbar segment, left side; standing position, lumbar segment, right side; prone position, thoracic segment, left side; prone position, thoracic segment, right side; prone position, lumbar segment, left side; prone position, lumbar segment, right side prone, position, trunk up, thoracic segment, left side; prone, position, trunk up, thoracic segment, right side; prone position, trunk up, lumbar segment, left side; prone position, trunk up, lumbar segment, right side; prone position, lower limbs up, thoracic segment, left side; prone position, lower limbs up, thoracic segment, right side; prone position, lower limbs up, lumbar segment, left side; prone position, lower limbs up, lumbar segment, right side.

The sEMG amplitude measurements were carried out according to the recommendations proposed by Surface ElectroMyoGraphy for the Non-Invasive Assessment of Muscles [17]. Both spinal testing with the Diers Formetric III 4D and erector spinae electromyographic (sEMG) examination were carried out in a non-invasive manner and without causing the subjects pain.

### Statistical Analysis

The graded, multiple progressive regression method was used to assess the correlation between erector spinae muscle tone (sEMG) and the concave and convex sides of the spinal curvature. The independent variables were erector spinae muscle tone evaluated via surface electromyography (sEMG amplitude). The dependent variables were assumed as convexity and concavity of the curvature. Prior to implementation of this model, the data set was divided by means of 10-fold cross-validation. The test and learning sets were created on the basis of the k subset. The stepwise multiple progressive regression model was introduced to the learning set. The test (validation) set was utilised for result reliability assessment. The results of 10-fold cross-validation were averaged into a single result. Statistical significance was verified using the F test, demonstrating the level of statistical significance for the standardised β regression coefficient. The quality of the fit was evaluated using the R^2^ determination coefficient. The level of statistically significant differences was assumed at *p* < 0.05.

## 3. Results

The study group comprised 103 participants (42.21%) suffering from scoliosis (IS) and 141 (57.79%) subjects demonstrating scoliotic posture (PS). In the group of participants with scoliotic posture (PS), curvature in the thoracic region was noted most frequently. Among the girls, this value totalled 62%, while for the boys it equalled 56%. In the group comprising participants demonstrating scoliotic posture (PS), thoracic curvatures were also observed, that is, for girls 49% and among boys 45%. Curvatures present on the left side were most common among the studied girls, both in the scoliosis group (41%) and in that with scoliotic posture (PS). In contrast, right-sided curvatures were more common among boys. The most frequently noted were single curvatures among girls from the scoliosis (IS) (77%) and scoliotic posture (PS) (63%) groups, while in the case of boys, in those from the group with scoliosis (IS) (56%) and scoliotic posture (PS) (52%). No triple curvatures were noted. The values obtained for location and dispersion measures, with regard to erector spinae SEMG amplitude, were of different distribution in both girls and boys among the scoliosis (IS) and scoliotic posture (PS) groups. In the case of girls, the largest absolute differentiation was noticed for the following variable under study: prone position, lumbar section, right side (group with scoliosis (IS)) (S = 47.58); (group exhibiting scoliotic posture (PS)) (S = 47.22). For boys, the highest absolute differentiation was also observed for the variable: prone position, lumbar segment, right side, and in both groups (scoliosis) (S = 51.33); (scoliotic posture) (PS) (S = 47.00) (see Table 1). In the examined girls, the highest generalised amplitude was noted for scoliosis (IS) (x = 47.17 mV) and the same was noted for boys (x = 48.72 mV) (see Table 1).

The stepwise regression model used to evaluate differences in erector spinae muscle tone between the thoracic region (sEMG amplification) and the convex and concave sides of the thoracic spine was very well-explained (R^2^ = 0.80; *p* = 0.001); (R^2^ = 0.89; *p* = 0.001) (Table 2). Regardless of position during examination in the thoracic spine, greater erector spinae (sEMG amplitude) tone occurred on the convex side of the curvature. This model was furthermore well-explained in the case of assessing differences between erector spinae muscle tone (sEMG amplitude) in the lumbar region and on the convex and concave sides of the lumbar curvature (R^2^ = 0.81; *p* = 0.001); (R^2^ = 0.62; *p* < 0.001) (Table 3). Regardless of position during examination in the lumbar region, greater erector spinae tone (sEMG amplitude) occurred on the curvature’s concave side, except in the case of the test carried out in standing position. In this test, a higher value of the tone was recorded on the convex side in the lumbar region.

## 4. Discussion

When considering concepts of idiopathic scoliosis (IS) formation as well as its development, the theory of muscle tension imbalances is of great significance. According to this theory, disturbances in the muscles are considered of secondary nature. Furthermore, they result from alterations occurring in the CNS that have not yet been investigated. In contrast, muscle disorders are primary in nature with regard to bone changes. Jointly, this results in a series of changes that make up the clinical and anatomical images of idiopathic scoliosis (IS). In various research trials, scientists highlight the crucial role of the nervous system in scoliosis pathogenesis. In the present study, the highest generalised electromyographic activity (sEMG amplitude) of the erector spinae was noted concerning scoliosis (IS), both among girls (x = 47.17 mV) and boys (x = 48.72 mV). The greater the curvature angle, the greater the muscle tone of the erector spinae. Small differences between scoliosis posture (PS) and scoliosis (IS) stem from the fact that the relationship between muscle tone of the erector spinae and the concave and convex sides of spinal curvature in low-grade scoliosis (IS) was investigated among children. Regardless of position during examination of the thoracic spine, greater tone (sEMG amplitude) of the erector spinae was exhibited on the convex side of the spinal curvature. However, in the lumbar, greater tone (sEMG amplitude) occurred on the concave side of the curvature. Differences in tone of the examined muscles in the thoracic and lumbar sections are related to the fact that, among the short muscles of the back, the most important is the muscle triad: multifaceted, long rotator, and short rotator muscles. Such triads constitute a cascade system of serially connected and overlapping motor units. In the lumbar region, the stretching, especially of the inter-transverse muscles, occurs on the convex side. The oblique course of these muscles in the thoracic section and the fact that the spinous processes move towards the convexities during a side bend cause the triad muscles on the concave side to be stretched. In this way, the imbalance is restored in both segments by the deep muscles at different ends, ipsilaterally in the thoracic region and contralaterally in the lumbar region. The modern approach to the problem of active spinal stabilisation has its source in the works by Kapadji [18], Bergmark [19], Cresswell [20], Panjabi [21], as well as Snijders et al. [22].

According to Roaf [23,24], idiopathic scoliosis (IS) is caused by muscle imbalance inducing spinal rotation. In Bayer’s [25] opinion, idiopathic scoliosis is due to asymmetrical work of the erector spinae muscles. This causes bioelectric activity to be larger on the convex side, and that is why the deformation occurs. Riddle and Roaf [26], Henssge [27], Butterworth and James [28], Redford et al. [29], and Alexander and Season [30] studied myoelectric activity in the case of an unloaded spine. In contrast, Güth et al. [31] and Güth and Abbink [32] assessed the EMG activity of gait assisted with and without braces. In their trials, Hertle and Jentschura [33], Le Febvre et al. [34], Brussatis [35], and Żuk [36] implemented different types of loaded positions. A noted increase in EMG amplitude and spontaneous activity on the convex side of the curvature, close to its apex, were viewed as the key research conclusions. 

Significant differences were noted for numerical EMG amplitude values between individuals. This was due to potential variation concerning muscle tone, body configuration, and muscle to geometry ratio of the electrode [37]. The moderately higher myoelectric amplitude observed for the signal in the paravertebral muscles on the convex side of the curvature, recorded in conditions of loading, is consistent with previously obtained findings [37]. The differences noted for both sides have been interpreted in various manners. The fatigue mechanism has been previously suggested as one of the potential interpretations [26,36]. In contrast, Butterworth and James [28] suggested that differences were visible due to the effects of erector spinae muscle stretching on the convex side. This interpretation can be further supported by noting a stretch reflex (H-reflex) being more sensitive to vibrations performed on the spinous processes in larger curvatures on the convex side [38,39]. The trunk buckles at axial loads as small as 20 N [40], i.e., far less than body segment masses above a specific level in the lumbar or mid-thoracic spinal segments [41,42]. Spinal stability is achieved via muscles. When scoliosis (IS) is present, a moment of flexion occurs, which acts on the spine within the area of the frontal plane and is directly proportional to the scoliosis (IS) degree. This indicates that, in order to assume and maintain upright postural positioning, the paraspinal muscles on the convex side need to be subjected to greater work than those present on the concave side. The results allow to suggest that the difference in EMG amplitude increases along with the increasing angle of scoliosis (IS). Amplitude of the myoelectric signal also rises along with the rise in exerted force [43]. In contrast, in accordance with the research conducted by Zetterberga et al. [44], it is suggested that the difference concerning EMG amplitude was not due to increased muscle activity on the convex side but was rather due to decreased concave side activity. In the case of this trial, there is no explanation as to why this occurs. The amplitude of myoelectric signal changed during the time of recording. However, this alteration was inconsistent for all electrode locations. Over time and at various lumbar levels, a decrease was noted in EMG. 

Nonetheless, at thoracic levels, an increase was visible with one exception, on the concave side in the scoliotic group. The results obtained for the indices of lumbar levels are consistent with the ones presented by Chapman and Troup [45]. In this study, participants within the norm were subjected to loads while assuming upright posture. However, many other researchers have noted increased amplitude of the signal during sustained contractions of the arm and hand muscles [46,47,48,49]. Nevertheless, the contraction level in the presented study was high. The increased values for EMG, observed at the L_3_ level (compared toT_8_ level), may be the result of applying a greater load at a lower level. The side-related variations of the EMG at the T_8_ level among the scoliosis group may also be due to applying a greater load on the convex side. Even so, other explanations for these differences should also be considered, i.e., variation in the length of muscle fibres, muscle area, as well as distance from active motor units to the electrodes [37,50]. The results obtained by Zetterberga et al. [44] allow to suggest that the loads placed on the convex and concave sides of the paraspinal muscles were in direct proportion to the capacity of these muscles. It seems that those on the convex side have adjusted to the demand of a greater load. Such a conclusion is in accordance with the findings of research on side-related differences in paravertebral muscle morphology [51].

The etiology of idiopathic scoliosis (IS) continues to be ambiguous. Nonetheless, a constant increasing allows to indicate spinal deformity as an expression of nervous system subclinical disorders. Defects in sensory input or anomalies in sensorimotor integration can induce irregularities in postural tone and, thus, the progression of spinal deformities. Inhibiting motor cortico-cortical excitability in dystonia is not considered normal. As a consequence, research on cortico-cortical inhibition may make the hypothesis of dystonia regarding IS pathophysiology more understandable.

Paired pulse transcranial magnetic stimulation was implemented to test cortico-cortical inhibition as well as facilitation in 9 young subjects experiencing idiopathic scoliosis, 5 adolescents demonstrating congenital scoliosis (CS), and 8 healthy age-matched controls. The effects of the previously applied conditioning stimulus (80% intensity regarding resting motor threshold) on motor-evoked potential amplitude was tested. This was induced by a test stimulus (120% of resting motor threshold) at differing inter-stimulus intervals (ISIs) in both muscles of the abductor pollicis brevis. The conclusions achieved for the healthy representatives and those with CS allowed to note a noticeable inhibitory consequence of the conditioning stimulus on the reaction to the test stimulus applied at inter-stimulus intervals (not lasting longer than 6 ms). These conclusions are identical to those reached for adult subjects being within the norm. Nonetheless, in the case of IS among children, abnormally decreased cortico-cortical inhibition at the short ISIs was observed. Cortico-cortical inhibition was virtually within the norm on the convex side of scoliosis, while being simultaneously and significantly decreased on its convex side. Concluding, the obtained findings allow to sustain the following hypothesis: dystonic dysfunction is indeed the underlying cause of IS. Asymmetrical cortical hyper-excitability may play a significant part in the pathogenesis of idiopathic scoliosis [52]. The limitation of the presented research was that testing was carried out only among a group with low-grade idiopathic scoliosis and with scoliotic posture. Therefore, we are planning an sEMG study of the back muscles in a group of children with scoliosis above 30°. In future research, we would also like to extend EMG tests to the short muscles of the back: musculus multifidus, long rotator, short rotator, as well as the inter-transverse.

## 5. Conclusions

The highest, generalised erector spinae tone (sEMG amplitude) was noted in the case of scoliosis. Regardless of position during examination of the thoracic spine, greater tone (sEMG amplitude) of the erector spinae muscles was exhibited on the convex side of the spinal curvature. However, in the lumbar spine, greater tone (sEMG amplitude) occurred on the concave side of the curvature. The exception was the test performed in standing position, during which greater tone was noted on the side of the convex side of the curvature. In therapeutic practice, within the thoracic section, too tense erector spinae muscles should be stretched on the convex side of the scoliosis, while in the lumbar region this should be performed on the concave side. However, each case of scoliosis should be treated individually. The presented research demonstrates great value as it is applicative in nature and allows to fill a research gap when it comes to erector spinae muscle tone in young children experiencing low-grade scoliosis. The development of scoliosis is associated with asymmetry and an increase in tone of the erector spinae. Its uneven tone occurring on both sides of the spine and in its various segments causes destabilisation and abnormal development. Our research may contribute to the creation of more effective diagnostic and therapeutic methods in the treatment of scoliosis (IS).

## Figures and Tables

**Table 1 children-08-01168-t001:** Muscle tone of the erector spinae (sEMG amplitude).

Variables	Girls				Boys			
	Scoliosis (IS)		Scoliotic Posture (PS)		Scoliosis (IS)		Scoliotic Posture (PS)	
	Mean mV	SD	Mean mV	SD	Mean mV	SD	Mean mV	SD
Position: standing; segment: thoracic; side: left	21.25	5.75	20.99	5.43	23.61	12.72	19.97	8.35
Position: standing; segment: thoracic; side: right	32.05	24.39	33.62	24.31	32.35	24.42	30.48	25.81
Position: standing; segment: lumbar; side: left	20.73	9.85	22.40	10.10	27.98	17.08	26.58	13.99
Position: standing; segment: lumbar; side: right	50.47	42.97	50.95	45.04	53.63	48.23	49.98	44.03
Position: prone; segment: thoracic; side: left	31.54	19.61	27.48	15.71	24.75	11.55	22.75	10.20
Position: prone; segment: thoracic; side: right	38.46	24.82	37.37	24.97	37.71	26.31	31.58	23.85
Position: prone; segment: lumbar; side: left	20.96	14.38	19.73	13.53	22.39	21.51	17.72	16.82
Position: prone; segment: lumbar; side: right	46.83	47.58	46.67	47.22	47.84	51.33	41.92	47.00

**Table 2 children-08-01168-t002:** Muscle tone of the erector spinae (sEMG amplitude).

Variables	Girls				Boys			
	Scoliosis		Scoliotic Posture		Scoliosis		Scoliotic Posture	
	Mean mV	SD	Mean mV	SD	Mean mV	SD	Mean mV	SD
Prone, position, trunk up, thoracic segment, left side	53.11	27.59	51.24	26.34	56.01	31.07	53.73	25.91
Prone, position, trunk up, thoracic segment, right side	68.83	22.61	65.70	21.12	67.76	29.11	69.65	23.73
Prone position, trunk up, lumbar segment, left side	52.53	27.07	48.36	22.52	55.36	28.82	51.89	24.76
Prone position, trunk up, lumbar segment, right side	76.65	41.29	73.24	34.78	78.55	39.26	74.08	33.65
Prone position, lower limbs up, thoracic segment, left side	36.42	18.68	34.08	17.07	38.60	23.08	40.55	24.20
Prone position, lower limbs up, thoracic segment, right side	47.05	18.45	49.07	25.24	53.61	34.41	56.05	33.04
Prone position, lower limbs up, lumbar segment, left side	63.44	34.21	63.50	37.75	64.47	34.76	62.73	30.46
Prone position, lower limbs up, lumbar segment, right side	94.48	47.20	85.32	36.30	95.00	42.18	91.86	32.39
Total	47.17	26.65	45.60	25.46	48.72	29.74	46.34	26.13

**Table 3 children-08-01168-t003:** Correlation between erector spinae muscle tone (sEMG) and the convex as well as concave sides of spinal curvature.

Variables			Thoracic Segment	Lumbar Segment
			Risk Factor β	Risk Factor β
Standing position	Convex side of curvature	Side of spine: left	0.5	0.18 *
		Side of spine: right	0.10 *	0.40 ***
	Concave side of curvature	Side of spine: left	0.01	0.21 **
		Side of spine: right	0.09	0.36 ***
Prone position	Convex side of curvature	Side of spine: left	0.61 ***	0.45 ***
		Side of spine: right	0.52 ***	0.25 **
	Concave side of curvature	Side of spine: left	0.55 ***	0.49 ***
		Side of spine: right	0.25 **	0.66 ***
Prone position, trunk up	Convex side of curvature	Side of spine: left	0.43 ***	0.002
		Side of spine: right	0.51 ***	0.01
	Concave side of curvature	Side of spine: left	0.22 **	0.12*
		Side of spine: right	0.38 ***	0.25 **
Prone position, lower limbs up	Convex side of curvature	Side of spine: left	0.21 **	0.19 *
		Side of spine: right	0.11 *	0.01
	Concave side of curvature	Side of spine: left	0.14 *	0.33 **
		Side of spine: right	0.02	0.05
Convex side of curvature			R^2^ = 0.80; *p* < 0.001	R^2^ = 0.81; *p* < 0.001
Concave side of curvature			R^2^ = 0.89; *p* < 0.001	R^2^ = 0.62; *p* < 0.001

* *p* < 0.05, ** *p* < 0.01, *** *p* < 0.001.

## Data Availability

Most of the relevant data are presented in the manuscript. The source data are archived in Pos-turology Laboratory, Collegium Medicum, Jan Kochanowski University.

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
