# Peer review of "Relationship between Muscle Tone of the Erector Spinae and the Concave and Convex Sides of Spinal Curvature in Low-Grade Scoliosis among Children"

_children, 2021, doi:10.3390/children8121168_

Round 1
Reviewer 1 Report
Prof Wilczynski presents and interesting study on the relationship between erector muscle tone and the potential influences on scoliosis. I think that the study idea is sound and attempts to answer a frequently asked question in the pathogensis of idiopathic scoliosis. I did have some questions and concerns regarding this paper:
1) I did find some of the grammar and sentence structure a little challenging. There were also a few spelling / syntactical errors that I think if corrected, would help improve the overall readability of the paper.
2) Table 1. is a little confusing and I think would be better suited if it could be summarized or presented in a more straightforward fashion.
3) Table 2. has a number of * that I would assume were to be explained in a table legend, but that has not been included.
4) I am curious as to why the differences in erector tone appear to be similar in patients with both scoiliosis and presumed postural scoliosis - I would have expected the pathophysiology of scoliosis in these patients to be different. Might the author have any thoughts or insight into this finding?
5) I am curious if the author has any suggestion as to the reasoning for why tone elevation occurs on different sides of the curve depending on whether it is thoracic or lumbar?
Author Response
Responses to the Reviewer’s comments
- Prof Wilczynski presents and interesting study on the relationship between erector muscle tone and the potential influences on scoliosis. I think that the study idea is sound and attempts to answer a frequently asked question in the pathogenesis of idiopathic scoliosis. I did have some questions and concerns regarding this paper:
- Point 1: I did find some of the grammar and sentence structure a little challenging. There were also a few spelling / syntactical errors that I think if corrected, would help improve the overall readability of the paper.
- Response 1: The manuscript has been proof-edited by a native speaker, professional translator and proof-editor: AmE Native, Katarzyna Smith-Nowak, ul. Miejscowa 8, 30-499 Kraków, e-mail: english.native123@gmail.com, Phone: 0048 505 990 391, NIP: PL 6792878140.
- Proof-editor: Katarzyna Smith-Nowak English native speaker (AmE), English Philology, M.A., Translation Studies, M.A., active translator/proof-editor.
- The text has been extensively and carefully proof-edited to ensure proper cohesion, coherence and style of the manuscript, while maintaining its substantive content. The general readability for an English-speaking audience should be appropriate, the flow of the text ensured. Below, please find a point-by-point list regarding the types of alterations/items checked in the text:
- Extensive stylistic editing (e.g. shortening sentences, dividing longer ones into two or more separate sentences, changes in word order, etc.),
- Eradication of repetitions and redundancies as well as spelling mistakes,
- Correction of grammatical number,
- Correction of spacing, punctuation,
- Exchanging colloquial linguistic items for more formal ones,
- Maintaining consistency of spelling version (British English),
- Correction of preposition usage,
- Correction of article usage,
- Maintaining linguistic consistency (e.g. medical linguistic items) within the text,
- Maintaining consistent usage of numerical format in data presentation,
- Corrections regarding parts of speech,
- Maintaining cohesion and coherence.
- Point 2:
Table 1. is a little confusing and I think would be better suited if it could be summarized or presented in a morstraightforward fashion.
- Response 2:
- In the ‘Material and Methods’ section, I have completed the description of the sEMG amplitude test procedure, and Table 1 is now clearer. The sEMG amplitude was recorded for the erector spinae on both sides, in the thoracic and lumbar regions - at the curvature arch apex and in the following positions: habitual standing; resting: prone (lower limbs extended in the knee joints, upper limbs placed along the trunk); under isometric contraction: in prone position, the subject lifted his/her trunk within the lumbar spine (approx. 30°), maintaining this position for 10 s, in prone position, upper trunk stabilised, the subject lifted both lower limbs within hip joint mobility (approx. 15°) for 10 s. A total of 16 SEMG amplitude measurements were taken for each child: standing position, thoracic segment, left side; standing position, thoracic segment, right side; standing position lumbar segment, left side; standing position, lumbar segment, right side; prone position, thoracic segment, left side; prone position, thoracic segment, right side; prone position, lumbar segment, left side; prone position, lumbar segment, right side prone, position, trunk up, thoracic segment, left side; prone, position, trunk up, thoracic segment, right side; prone position, trunk up, lumbar segment, left side; prone position, trunk up, lumbar segment, right side; prone position, lower limbs up, thoracic segment, left side; prone position, lower limbs up, thoracic segment, right side; prone position, lower limbs up, lumbar segment, left side; prone position, lower limbs up, lumbar segment, right side.
- Point 3: Table 2. has a number of * that I would assume were to be explained in a table legend, but that has not been included.
- Response 3: The number of asterisks (*) by risk factor β means: * is the level of statistical significance * p<0.05; **p<0.01, ***p<0.001; e. significance of differences in erector spinae tone between the concave and the convex sides of the spinal curvature. The more asterisks, the greater the difference.
- Point 4: I am curious as to why the differences in erector tone appear to be similar in patients with both scoiliosis and presumed postural scoliosis - I would have expected the pathophysiology of scoliosis in these patients to be different. Might the author have any thoughts or insight into this finding?.
- Response 4: In the present study, the highest generalised electromyographic activity (sEMG amplitude) of the erector spinae was noted in the case of scoliosis, both among girls (x=47.17 mV) and boys (x=48.72 mV). The greater the curvature angle, the greater the muscle tone of the erector spinae. Small differences between scoliosis posture and scoliosis stem from the fact that the relationship between muscle tone of the erector spinae and the concave and convex sides of spinal curvature in low-grade scoliosis was investigated among children. In future research, sEMG testing should be carried out among a group of children with severe scoliosis, i.e. that exceeding 30°.
- Point 5: I am curious if the author has any suggestion as to the reasoning for why tone elevation occurs on different sides of the curve depending on whether it is thoracic or lumbar?
- Response 5: Differences in tone of the examined muscles in the thoracic and lumbar sections are related to the fact that among the short muscles of the back, the most important is the muscle triad: multifaceted (in Latin: musculus multifidus), long rotator (Latin musculi rotatores long) and short rotator (Latin musculi rotatores short) muscles. Such triads constitute a cascade system of serially connected and overlapping motor units. In the lumbar region, the stretching, especially of the inter-transverse muscles (in Latin: intertransversus), occurs on the convex side. The oblique course of these muscles in the thoracic section and the fact that the spinous processes move towards the convexities during a side bend cause the triad muscles on the concave side to be stretched. In this way, the imbalance is restored in both segmentsby the deep muscles at different ends, ipsilaterally in the thoracic region and contralaterally in the lumbar region.
- The modern approach to the problem of active spinal stabilisation has its source in the works by Kapadji [19], Bergmark [20], Cresswell [21], Panjabi [22], as well as Snijders et al [23].The deep back muscles play a special role in stabilising the spine. The deep back muscles are called the erector spinae.In humans, the primary segmental arrangement of muscle fibres is preserved in the deepest layers of the back. On the other hand, in more superficial layers, the muscle stramds become longer and cover a larger number of vertebrae in their course. The course and attachments of the muscle fibres allow to distinguish the following systems: transverse-spinous, interspinous and inter-transverse. The transverse-spinous and interspinous systems form the so-called medial strand, while the lateral strand is formed by the inter-transverse muscles (intertransversarii) and the erector spinae. The erector spinae is divided into the iliocostal muscle (iliocostalis) and the longest muscle (longissimus).The following muscles also play a significant part in stabilising the trunk: the romboid (m. rhomboideus), the serratus anterior, the broadest back muscle (latissimus dorsi), the trapezius dorsi and the quadratus lumborum), as well as the greater pectoral (pectorialis major), smaller pectoral (pectorialis minor) and suboccipital (suboccipitales). Summarising, it can be stated that the correct shape of the spine is maintained by the deep back muscles, which work with the muscles of the neck, the superficial muscles of the back, chest, abdomen, buttocks and lower limbs.The role of the short back muscles is the most complex. The intercostal and interspinous muscles are of great importance, as spinal stabilisers, although they are subject to our will, most often work when they are stimulated by stretching. This means that as a result of the action of muscles, one of the attachments is outside the spine, and they are activated to restore the unbalanced spine. Among the short muscles of the back, the following also play an important role: the multifidus, long (rotatores longi) and short rotator (rotatores breves). Such triads constitute a cascade system of serially connected and overlapping motor units. In the lumbar region, stretching, especially of the transverse muscles, occurs on the convex side. The oblique course of these muscles in the thoracic section and the fact that the spinous processes move towards the convexities during lateral bend cause the triad muscles on the concave side to be stretched. In this way, imbalance is restored in both segments by the deep muscles at different ends, ipsilaterally in the thoracic region and contralaterally in the lumbar region.In selecting corrective methods, it should be borne in mind that there are certain functional relationships between the above-mentioned reference systems, in the form of complete inter-system syntony, synkinesia and alternating balance, as well as functional antagonism of the deep muscles against the superficial spine. Anatomy trains are assigned a role for the statics of the body, including postural stabilisation, which further includes a set of muscles connected by fascia. It is also important that there are proprioreceptors in the fascia. As a result, the body does not function "in parts", but forms a functional whole. Additionally, in the myofascial, so-called trigger points are present. These play a significant role in the generation of pain and related movement blockages. The following trains can be distinguished: superficial (anterior, posterior and lateral), limb (upper and lower), spiral, functional and deep anterior. Each of the trains performs a different function in movement, and the increase in tone within one train leads to self-stabilisation, because e.g. an increase in tensile stress within the tendons and fascia is counterbalanced by an increase in compressive stress within the bone. This phenomenon is called tensegration. Disorders of the above-type tensions lead to both postural changes and impairment of movement patterns. A special role is assigned to the spiral and superficial back trains. The first one protects the trunk and lower limbs against the effects of torsional forces. It also helps to maintain balance in all planes, and in the event of imbalances, it participates in the control of compensatory movements. The function of the latter is to keep the body fully extended and prevent assuming a flexed position.
- I hope that our detailed responses and the extensive changes to the text are sufficient for the publication of our text. Thank you for your devoting you time and effort.
Yours sincerely,
Assoc. Prof. UJK Jacek Wilczyński, Ph.D.
Reviewer 2 Report
The idea of ​​the paper is interesting, but the execution does not consider some more current concepts about the etiology and development of idopathic scoliosis. In the Introduction, the authors begin citations with article 8.!
In the methodology, the authors need to make it clear which type of scoliosis they are analyzing, whether juvenile idiopathic or simply postural, and clearly separate the groups.
The authors chose the surface electromyography method to study the erector spinal muscles, but more recent studies show the involvement of multifidus in the etiology of idiopathic scoliosis. Are the authors able, by the method used, to differentiate the erector spinal muscles from the multifidus?
In the discussion, the authors use around 30 articles, only 1 (48) from 2010, the others are from the 20th century, including 1 article from 1923, with important historical value, but the authors need to use more recent articles, because the concepts of scoliosis and its etiology have come a long way in the last 10 years and should be considered.
The authors need to be more critical of the limitations of the study, making clear the limitations of surface electromyography.
The conclusions obtained are simplistic and do not consider the quality of the affected muscles and the possibility that they may have myopathy. The authors also need to elucidate the concept of primary and secondary curves, which can help explain the different behavior of the thoracic and lumbar curves.
Therefore, I suggest that the authors clarify the studied groups, update the concepts on the etiology of scoliosis and clarify the limitations of the method used.
Author Response
Responses to the Reviewer’s comments
- Point 1: The idea of ​​the paper is interesting, but the execution does not consider some more current concepts about the etiology and development of idopathic scoliosis. In the Introduction, the authors begin citations with article 8.!
- Response 1: The citations 1-7 in the article have been corrected, I apologize for the error.
- Point 2: In the methodology, the authors need to make it clear which type of scoliosis they are analyzing, whether juvenile idiopathic or simply postural, and clearly separate the groups.
- Response 2:
- The subjects were clearly divided into 2 groups: children with low-grade idiopathic scoliosis and scoliotic posture were examined. In accordance with the guidelines provided by the manufacturer of the Diers Gormetric III 4D device, scoliosis and/or scoliotic posture are identified by taking the values of 3 variables into consideration: pelvic tilt expressed (/millimetres), lateral deviation (/millimetres), as well as and surface rotation (/degrees). Scoliotic posture occurs in the case of pelvic tilt totalling 1-4 mm, while lateral deviation must be between 1-4 mm, and surface angle equalling between 1-4°. Scoliosis is present when pelvic tilt and lateral deviation are equal to or is greater than 5 mm, while surface rotation totals or is greater than 5°. In cases for which these 3 requirements were not met, it was assumed that neither scoliosis nor scoliotic posture were present [16].
- Point 3: The authors chose the surface electromyography method to study the erector spinal muscles, but more recent studies show the involvement of multifidus in the etiology of idiopathic scoliosis. Are the authors able, by the method used, to differentiate the erector spinal muscles from the multifidus ?
- Response 3: The aim of the current study was to assess the relationship between muscle tone of the erector spinae and the concave and convex sides of spinal curvature in low-grade scoliosis among children. The sEMG examination does not allow for direct examination of multifidus muscle tone, which would require the use of deep EMB with electrode insertion. Neighbouring muscles can give off an EMG signal, which is detected by the electrode. Usually, the "Cross Talk" phenomenon does not exceed 10-15% of the total signal content or does not occur at all. Care should be taken when placing the electrodes narrowly within the muscle groups. The modern approach to the problem of active spinal stabilisation has its source in the works by Kapadji [1], Bergmark [18], Cresswell [1], Panjabi [22-24], as well as Snijders et al [1 ].The deep back muscles play a special role in stabilising the spine. The deep back muscles are called the erector spinae.In humans, the primary segmental arrangement of muscle fibres is preserved in the deepest layers of the back. On the other hand, in more superficial layers, the muscle stramds become longer and cover a larger number of vertebrae in their course. The course and attachments of the muscle fibres allow to distinguish the following systems: transverse-spinous, interspinous and inter-transverse. The transverse-spinous and interspinous systems form the so-called medial strand, while the lateral strand is formed by the inter-transverse muscles (intertransversarii) and the erector spinae. The erector spinae is divided into the iliocostal muscle (iliocostalis) and the longest muscle (longissimus).The following muscles also play a significant part in stabilising the trunk: the romboid (m. rhomboideus), the serratus anterior, the broadest back muscle (latissimus dorsi), the trapezius dorsi and the quadratus lumborum), as well as the greater pectoral (pectorialis major), smaller pectoral (pectorialis minor) and suboccipital (suboccipitales). Summarising, it can be stated that the correct shape of the spine is maintained by the deep back muscles, which work with the muscles of the neck, the superficial muscles of the back, chest, abdomen, buttocks and lower limbs.The role of the short back muscles is the most complex. The intercostal and interspinous muscles are of great importance, as spinal stabilisers, although they are subject to our will, most often work when they are stimulated by stretching. This means that as a result of the action of muscles, one of the attachments is outside the spine, and they are activated to restore the unbalanced spine. Among the short muscles of the back, the following also play an important role: the multifidus, long (rotatores longi) and short rotator (rotatores breves). Such triads constitute a cascade system of serially connected and overlapping motor units. In the lumbar region, stretching, especially of the transverse muscles, occurs on the convex side. The oblique course of these muscles in the thoracic section and the fact that the spinous processes move towards the convexities during lateral bend cause the triad muscles on the concave side to be stretched. In this way, imbalance is restored in both segments by the deep muscles at different ends, ipsilaterally in the thoracic region and contralaterally in the lumbar region.In selecting corrective methods, it should be borne in mind that there are certain functional relationships between the above-mentioned reference systems, in the form of complete inter-system syntony, synkinesia and alternating balance, as well as functional antagonism of the deep muscles against the superficial spine. Anatomy trains are assigned a role for the statics of the body, including postural stabilisation, which further includes a set of muscles connected by fascia. It is also important that there are proprioreceptors in the fascia. As a result, the body does not function "in parts", but forms a functional whole. Additionally, in the myofascial, so-called trigger points are present. These play a significant role in the generation of pain and related movement blockages. The following trains can be distinguished: superficial (anterior, posterior and lateral), limb (upper and lower), spiral, functional and deep anterior. Each of the trains performs a different function in movement, and the increase in tone within one train leads to self-stabilisation, because e.g. an increase in tensile stress within the tendons and fascia is counterbalanced by an increase in compressive stress within the bone. This phenomenon is called tensegration. Disorders of the above-type tensions lead to both postural changes and impairment of movement patterns. A special role is assigned to the spiral and superficial back trains. The first one protects the trunk and lower limbs against the effects of torsional forces. It also helps to maintain balance in all planes, and in the event of imbalances, it participates in the control of compensatory movements. The function of the latter is to keep the body fully extended and prevent assuming a flexed position.
- Our goal was not to determine the tone (sEMG amplitude) of the multifidus muscles. Due to their large role in the development of scoliosis, we plan to do such research in the future.
- Point 3: In the discussion, the authors use around 30 articles, only 1 (48) from 2010, the others are from the 20th century, including 1 article from 1923, with important historical value, but the authors need to use more recent articles, because the concepts of scoliosis and its etiology have come a long way in the last 10 years and should be considered.
- Response 3:
- We wanted to present EMG research against a historical background. We supplemented the discussion with the latest works: Kwok, G.; Yip, J.; Cheung, M.C.; Yick K.L. Evaluation of Myoelectric Activity of Paraspinal Muscles in Adolescents with Idiopathic Scoliosis during Habitual Standing and Sitting. Biomed Res Int. 2015, 2015, 958450. doi: 10.1155/2015/958450.
- Point 4: The authors need to be more critical of the limitations of the study, making clear the limitations of surface electromyography.
- Response 4: The sEMG test has its advantages. The main benefit is that it is non-invasive. Examination of deep EMG - through the insertion of the electrode, in more than 200, 7-8-year-old children, with low-grade curvatures, is practically impossible for many reasons. The aim of the study was to assess the relationship between the muscle tone of erector spinae and the concave and convex sides of spinal curvature in low-grade scoliosis among children. It was assumed that there was a clear relationship between erector spinae muscle tone and the concave as well as convex sides of the spinal curvature in low-grade scoliosis among children. In the thoracic spine, greater erector spinae tone occurs on the convex side of the curvature. On the other hand, in the lumbar region, greater tone is noted on the concave side.
- Point 5: The conclusions obtained are simplistic and do not consider the quality of the affected muscles and the possibility that they may have myopathy. The authors also need to elucidate the concept of primary and secondary curves, which can help explain the different behavior of the thoracic and lumbar curves.
- Response 5:
- The children under study demonstrated low-grade idiopathic curvature and scoliotic posture. The inclusion criteria for participation in the study were presence of scoliosis, scoliotic posture, age between 7 and 8 years, parents' or legal guardians' written informed consent to perform the examination. The exclusion criteria for participation were syndromes related to the CNS and/or locomotor system preventing adequate psychomotor development (including myopathy) disorders, potentially causing pathological posture: genetic syndromes, hormone-based disorders, neuromuscular disease (including myopathy), congenital motor system defects, or lack of written consent to take part in the test protocol. The children and their parents/guardians were provided with information regarding the objective of the study, its course and duration. The terms primary and secondary curvature were also explained. In the case of multi-curvatures, one of them is primary (curvatura primaria). A primary curvature is in the shape of an arch that is regular and equal in both of its halves. The primary curvature is always larger and the most permanent, and therefore, the least susceptible to correction.. The primary curvature is, in most cases, compensated by a secondary curvature of the opposite direction. Secondary curvature (curvatura secudariae) is a symptom of striving to compensate for disturbances in the mechanical axis of the spine. While primary curvature is a negative factor, the secondary one, restoring balance and statics of the trunk, although under changed conditions, should be treated as a positive phenomenon. The attempt to compensate for disturbances in body axis in the curvature of the spine by producing counter-curvatures or distortions of the pelvis is called compensation.
- Point 6: Therefore, I suggest that the authors clarify the studied groups, update the concepts on the etiology of scoliosis and clarify the limitations of the method used.
- Response 6:
- We have made the study groups more specific: Children with low-grade idiopathic scoliosis and scoliotic posture were examined. In accordance with the guidelines provided by the manufacturer of the Diers Gormetric III 4D device, scoliosis and/or scoliotic posture are identified by taking the values of 3 variables into consideration: pelvic tilt expressed (/millimetres), lateral deviation (/millimetres), as well as and surface rotation (/degrees). Scoliotic posture occurs in the case of pelvic tilt totalling 1-4 mm, while lateral deviation must be between 1-4 mm, and surface rotation equalling between 1-4 . Scoliosis is present when pelvic tilt and lateral deviation is equal to or greater than 5 mm, while surface rotation totals or is greater than 5°. In cases for which these 3 requirements were not met, it was assumed that neither scoliosis nor scoliotic posture were present [16].
- updating the concepts on the etiology of scoliosis: A number of researchers find the cause of scoliosis to be in bone structures, in the so-called passive apparatus of the spine. They highlight the significance of disturbances in the epiphysial cartilage of the spinal vertebrae [9]. Nonetheless, at present, the majority of researchers recognise the primary disturbances as being in the CNS, which lead to disturbances in muscle balance. Changes in the passive elements of the spine are, according to them, secondary [10]. In scoliosis, muscle balance of the main, active, spinal stabilisers, i.e. the erector spinae, can be subjected to disturbances caused by primary dysfunctions in the structures of the CNS [11]. This, as a result, leads to disturbances in erector spinae activity and tension, causing the development of curvatures. Much allows to conclude that the discrete functional changes in the CNS are the primary cause of idiopathic scoliosis [12]. Under the influence of the etiological factor, disturbance of internal spinal balance occurs. In the first phase, there is only a narrowing of the intervertebral discs on the concave side of the scoliosis and a loss of their elasticity. At the same time, an uneven an increase in erector spinae muscle tone (sEMG amplitude) can be observed. Both in the diagnosis and in preventative treatment of scoliosis, much attention is devoted to the evaluation of differences in erector spinae tone between the concave and convex side of the scoliosis and at various levels of the spine [13].
- and clarify the limitations of the method used: The limitation of the presented research was testing carried out only among a group with low-grade idiopathic scoliosis and with scoliotic posture. Therefore, we are planning an sEMG study of the back muscles in a group of children with scoliosis above 30°. In future research, we would also like to extend EMG tests to the short muscles of the back: musculus multifidus, long rotator, short rotator, as well as the intertransverse musucle.
I hope that our detailed responses and the extensive changes to the text are sufficient for the publication of our text. Thank you for your devoting you time and effort.
Yours sincerely,
Assoc. Prof. UJK Jacek Wilczyński, Ph.D.
Round 2
Reviewer 1 Report
Prof Wilcynski has resubmitted a revised version of their previously submitted manuscript. I think that some of my previous concerns have been addressed. I think there is an opportunity for refinement prior to publication.
1) Table 1. remains quite long and perhaps unwieldly - perhaps it would be worthwhile to break this table up?
2) Thank you for the clarification for the meaning of the * in Table 2. I would suggest placing this clarification in a legend for this table.
3) I think readability has been improved, but there remain what appear to syntactical mistakes. For example, on line 25 of the abstract, I think there is a word missing ("In therapeutic practice, this means that the on the convex side...). I find that the introduction especially has many instances of run on sentences which does impede the general readability of the text.
Author Response
Responses to the Reviewer’s comments
Prof Wilcynski has resubmitted a revised version of their previously submitted manuscript. I think that some of my previous concerns have been addressed. I think there is an opportunity for refinement prior to publication.
- Point 1: Table 1. remains quite long and perhaps unwieldly - perhaps it would be worthwhile to break this table up?
- Response 1: I have corrected the tables and divided them into 2 separate ones.
- Point 2: Thank you for the clarification for the meaning of the * in Table 2. I would suggest placing this clarification in a legend for this table.
- Response 2: * I have put an explanation in the legend to this table.
- Point 3: I think readability has been improved, but there remain what appear to syntactical mistakes. For example, on line 25 of the abstract, I think there is a word missing ("In therapeutic practice, this means that the on the convex side...). I find that the introduction especially has many instances of run on sentences which does impede the general readability of the text.
- Response 2: I have corrected the sentence: “In therapeutic practice, within the thoracic section, too tense erector spinae muscles should be stretched on the convex side of the scoliosis, while in the lumbar region, this should be done on the concave side”.
- I apologize for the mistakes made.
I hope that our responses and changes to the text are sufficient for the publication of our text. Thank you for devoting your time and effort.
Yours sincerely,
Assoc. Prof. UJK Jacek Wilczyński, Ph.D.
Reviewer 2 Report
I recognize the author's effort in correcting the paper, but the following questions remain:
1- You need to put the meaning of the acronym CNS logo in the Introduction
2 - The introduction quotes "In the treatment of scoliosis, there is an opinion that in order
to obtain a permanent corrective effect, regardless of the curvature location (thoracic,
lumbar region), the spinae erector on the concave side of the curvature should be
stretched while the muscles on the convex side should be strengthened". Whose opinion is this (which source consulted?)
3- Why did the authors choose the age between 7 and 8 years old? This choice is dubious, as it may include patients with resolving infantile idiopathic scoliosis, patients with juvenile idiopathic scoliosis (most likely) and/or patients with early onset of adolescent idiopathic scoliosis.
4- I suggest that the authors clarify the names of the groups, as calling them scoliosis and scoliotic posture causes confusion. The author himself gets confused right at the beginning of the presentation of the results! Thus, I suggest calling the groups idiopathic scoliosis, with abbreviation of (IS) and postural scoliosis, abbreviating as (PS).
5- I do not think it is appropriate for the author to include in the discussion his opinion about sEMG biofeedback, including that of Figure 1, as it changes the focus of the good work performed, the author even puts the following sentence, without citing the sources: "There is a number of research work in the literature that have applied sEMG biofeedback as
an instrument for muscle rehabilitation"!
Author Response
Responses to the Reviewer’s comments
- I recognize the author's effort in correcting the paper, but the following questions remain:
- Point 1: You need to put the meaning of the acronym CNS logo in the Introduction.
- Response 1: I have placed the meaning of the CNS acronym in the ‘Introduction’ section: Nonetheless, at present, the majority of researchers recognise the primary disturbances as being in the Central Nervous System (CNS), which lead to disturbances in muscle balance.
- Point 2: The introduction quotes "In the treatment of scoliosis, there is an opinion that in order to obtain a permanent corrective effect, regardless of the curvature location (thoracic, lumbar region), the spinae erector on the concave side of the curvature should be stretched while the muscles on the convex side should be strengthened". Whose opinion is this (which source consulted?).
- Response 2: I have deleted this sentence. I meant that: “Among physiotherapists dealing with the treatment of scoliosis, there is sometimes an opinion that in order to obtain a corrective effect, regardless of the location of the curvature (thoracic, lumbar), the erector spinae should be stretched on the concave side of the curvature, while the muscles on the convex side should be strengthened”.
- Point 3: Why did the authors choose the age between 7 and 8 years old? This choice is dubious, as it may include patients with resolving infantile idiopathic scoliosis, patients with juvenile idiopathic scoliosis (most likely) and/or patients with early onset of adolescent idiopathic scoliosis.
- Response 3: The selection of children aged 7-8 for the study was related to the fact that the first critical period of posturogenesis occurs in the initial school period. Between the ages of 7-8, the lifestyle of children changes. The essence of this change consists in the transition from a free (individually regulated movement by the child - effort and rest), to an imposed system comprising several hours of sitting, sometimes in inappropriate conditions. The second critical period of posturogenesis is related to the pubertal growth spurt that occurs in girls between the age of 11-12, while in boys, between the age of 13-14. The first scoliotic changes appear most often in these stages.
- Point 4: I suggest that the authors clarify the names of the groups, as calling them scoliosis and scoliotic posture causes confusion. The author himself gets confused right at the beginning of the presentation of the results! Thus, I suggest calling the groups idiopathic scoliosis, with abbreviation of (IS) and postural scoliosis, abbreviating as (PS).
- Response 4: Throughout the article, I used the abbreviation (SI) for scoliosis and (PS) for scoliotic posture.
- Point 5: I do not think it is appropriate for the author to include in the discussion his opinion about sEMG biofeedback, including that of Figure 1, as it changes the focus of the good work performed, the author even puts the following sentence, without citing the sources: "There is a number of research work in the literature that have applied sEMG biofeedback as an instrument for muscle rehabilitation”!
- Response 5: I have deleted the information about sEMG biofeedback as well as Figure 1 from the ‘Discussion’ section.
- I apologize for the mistakes made.
I hope that our responses and changes to the text are sufficient for the publication of our text. Thank you for devoting your time and effort.
Yours sincerely,
Assoc. Prof. UJK Jacek Wilczyński, Ph.D.
Round 3
Reviewer 2 Report
Thank you for sending the paper and for responding to my suggestions. I am satisfied with the changes made.